# Effects of Different Amounts of Corn Silk Polysaccharide on the Structure and Function of Peanut Protein Isolate Glycosylation Products

**DOI:** 10.3390/foods11152214

**Published:** 2022-07-26

**Authors:** Xinxin Han, Yilin Zhao, Sining Mao, Nannan Hu, Dijing Sun, Qi Yang, Zejun Chu, Qihang Zheng, Lin Xiu, Jingsheng Liu

**Affiliations:** 1College of Food Science and Engineering, Jilin Agricultural University, Changchun 130118, China; hanxinxin0519@126.com (X.H.); jllyzyl@126.com (Y.Z.); maosining@126.com (S.M.); hunannan@126.com (N.H.); sundijing@126.com (D.S.); yangqi@126.com (Q.Y.); chuzejun@126.com (Z.C.); zhengqihang@126.com (Q.Z.); liujingsheng@jlau.edu.cn (J.L.); 2National Engineering Research Center for Wheat and Corn Deep Processing, Changchun 130118, China

**Keywords:** peanut protein isolate, glycosylation, corn silk polysaccharide, functional properties

## Abstract

Covalent complexes of peanut protein isolate (PPI) and corn silk polysaccharide (CSP) (PPI-CSP) were prepared using an ultrasonic-assisted moist heat method to improve the functional properties of peanut protein isolate. The properties of the complexes were affected by the level of corn silk polysaccharide. By increasing the polysaccharide addition, the grafting degree first increased, and then tended to be flat (the highest was 38.85%); the foaming, foam stability, and solubility were also significantly improved. In a neutral buffer, the solubility of the sample with a protein/polysaccharide ratio of 2:1 was 73.69%, which was 1.61 times higher than that of PPI. As compared with PPI, the complexes had higher thermal stability and lower surface hydrophobicity. High addition of CSP could made the secondary structure of PPI change from ordered *α*-helix to disordered *β*-sheet, *β*-turn, and random coil structure, and the complex conformation become more flexible and loose. The results of multiple light scattering showed that the composite solution exhibited high stability, which could be beneficial to industrial processing, storage, and transportation. Therefore, the functional properties of peanut protein isolate glycosylation products could be regulated by controlling the amount of polysaccharide added.

## 1. Introduction

Peanut is one of the most important oil crops in the world. Since 1993, China’s total peanut output has ranked first in the world. Peanuts are mainly used to produce oil products, and the remaining peanut meal from oil extraction is used to process animal feed, which contains high levels of peanut protein isolate (PPI), resulting in a significant waste of protein resources. Peanut protein contains essential amino acids [1] and has high nutritional value [2]. However, as compared with other plant proteins, its functionality is poor (such as solubility and thermal stability), which greatly limits the application of peanut protein [3].

Corn silk polysaccharide is a polysaccharide obtained from corn silk, mainly composed of glucose, galactose, arabinose, galacturonic acid, mannan, and xylose [4]. Studies have shown that polysaccharide, as an important active ingredient in corn filaments, has unique pharmacological effects and various biological activities such as antioxidant, anti-diabetic, and hypolipidemic, and therefore it has attracted more and more attention [5]. Corn silk is usually treated as a waste in food processing and has not received enough attention.

Improving the characteristics of protein is an important area of research. Existing studies have typically used physical, chemical, and enzymatic treatments to improve the functional activity of PPI [6,7,8], but may have introduced some compounds that are harmful to human health. Glycosylation does not require any catalyst, and is a safe and non-polluting method for protein modification, having great potential in the food industry. The glycosylation of proteins is based on the principle of the Maillard reaction. A Schiff base is formed by the condensation reaction between the aprotonated amino group of a protein and the reduced carbonyl group of the polysaccharide molecule, and then the rearrangement product of the glycosyl amine is generated, which can significantly improve the functional properties of the protein [9]. Complexes formed by protein and polysaccharide molecules through glycosylation can improve the solubility, foamability, and other functional properties of proteins. Currently, there are many studies on glycosylation to improve the functional properties of proteins. The glycosylation of soybean protein isolate has been shown to significantly improve the solubility and emulsification of complexes [10,11]. Preparation of a *β*-conglycinin (7S)-MD covalent complex (7S-MDUH) using an ultrasonic method combined with a moist heat method Maillard reaction has been performed which improved the surface hydrophobicity, emulsification, and emulsion stability of the covalent complexes [12]. The Maillard reaction of oat protein isolate and glucan has been shown to increase the solubility of protein and significantly improve the emulsification and foaming properties of oat protein isolate [13]. Bouyer, Eléonore, and Mekhloufi [14] studied the foaming properties of *β*-lactoglobulin-galactose covalent complexes under different pH conditions. Glycosylation had little effect on the interfacial properties and foaming at pH = 7, however, at pH = 5, the covalent complexes exhibited better kinetic adsorption at the air–water interface and had better foaming properties as compared with a control group. Although glycosylation has been extensively studied in protein modification, the structures and functions of peanut protein isolate and corn silk polysaccharide (CSP) glycosylation products have not been reported.

In this study, the glycosylation products of PPI-CSP are prepared using an ultrasonic-assisted moist heat method. The functional properties of the complexes are regulated by controlling the amount of polysaccharide added, and we characterize the structures and properties of the complexes. The aim is to explore the effect of different amounts of CSP on the structures and functions of PPI glycosylation products, and to provide a theoretical basis for improving the functional properties of peanut protein isolate and its application in the food industry.

## 2. Materials and Methods

### 2.1. Materials and Chemicals

The corn silk polysaccharide was extracted from Kennuo No. 1 corn silk, (methods for extraction, purification, and purity determination of corn silk polysaccharide are described in Appendix B). The purity was 83.75 ± 2.38%. The structural characterization of corn silk polysaccharide and other analysis references are available in the Appendix C. 98% peanut protein isolate (Bellancom Distributor of China, Beijing, China) and o-phthalaldehyde (OPA) (Sigma-Aldrich, St. Louis, MA, USA). Sodium tetraborate, and 1-anilino-8-naphtalene-sulfonate (ANS) were purchased from Yuanye Biotechnology Co. Ltd. (Shanghai, China). Bovine serum albumin (BSA) and *β*-mercaptoethanol were purchased from Aladdin Biochemical Technology Co. Ltd. (Shanghai, China). All other chemicals used in this work were analytical grade.

### 2.2. Preparation of PPI-CSP Complexes

The PPI-CSP complexes were based on the method by Zhao et al. [15], with minor modifications. PPI (1 g) was dissolved in 0.1 L of 0.1 mol/L pH = 7 phosphate buffer (PB) and magnetically stirred for 1 h to obtain the PPI solution (1%, *w*/*v*). Then, CSP (0.1, 0.2, 0.5, 1, and 2 g) was added to the PPI solution to achieve the following mass PPI/CSP ratios: 10:1, 5:1, 2:1, 1:1, and 1:2, respectively. The PPI-CSP mixtures were then stirred for 1 h at room temperature and stored at 4 °C overnight to ensure complete dissolution and homogeneity. The samples were placed in a sonicator (JY92-2D, NingBoScientz Biotechnology Co. Ltd., Ningbo, China) under ice bath conditions, and a 0.54 cm ultrasonic probe was used to sonicate the samples. Sonication conditions were pulse duration 2 s, off time 2 s, and output power 480 W for 60 min. The sonicated samples were placed in a 95 °C water bath and heated for 30 min. Immediately after the heat treatment, the samples were placed in an ice-water bath and cooled to room temperature. The final complex solutions were lyophilized and stored at 4 °C for further analysis. The PPI with different CSP mass ratios were labeled as PC10:1, PC5:1, PC2:1, PC1:1,and PC1:2, respectively. The untreated peanut protein isolate was labeled as PPI and used as a control.

### 2.3. Degree of Graft (DG)

Free amino groups were determined by an o-phthaldialdehyde assay [16]. First, 80 mg of OPA was dissolved in 2 mL of methanol and mixed with 50 mL of 100 mM sodium tetraborate, 5 mL of 20% (*w*/*w*) SDS, and 200 μL of *β*-mercaptoethanol. The OPA reagent was prepared by diluting the mixed solution to 100 mL with distilled water. Then, 200 μL of complex solutions (2 mg/mL) were incubated with 4 mL of OPA reagent at 90 °C for 5 min. The absorbance at 340 nm was measured by using an ultraviolet-visible spectrophotometer (UNICO UV-2100, Shanghai, China) to obtain the free amino group content. L-serine was used as a standard.
(1)$$\mathrm{Degree}\;\mathrm{of}\;\mathrm{graft}\;\left(\mathrm{DG}\right)=\frac{A_{0}-A_{t}}{A}\times 100$$
where the relative level of free amino groups is expressed by absorbance. A_0_ is the absorbance of the PPI-CSP mixture, A_t_ is the absorbance of the PPI-CSP complexes, and A is the absorbance of the PPI solution.

### 2.4. Measurement of Surface Hydrophobicity (H_0_)

The measurement of surface hydrophobicity was based on methods reported by Chandrapala et al. [17] and Benoît et al. [18] with slight modifications. The fluorescent probe 1-anilino-8-naphthalenesulfonate (ANS) was used to label the hydrophobic groups on the surface of the sample, determine the H_0_ of the complexes, and dilute the complexes with 10 mM PB to obtain 0.0625, 0.125, 0.25, 0.5, and 1 mg/mL complex solutions. Then, 20 μL of 8 mmol/L ANS solution was added to 2 mL of sample solution, mixed, and incubated at 25 °C for 3 min. The sample (200 μL) was placed in a 96-well microtiter plate and measured at 390 nm (excitation) and 470 nm (emission) using a FLUO star Omega multifunctional microplate reader (BMG LABTECH, Ortenberg, Germany), both with the same slit width of 5 nm. Curves with the protein concentration as the abscissa and the FI value as the ordinate were established, and the initial slopes of these curves were used as the surface hydrophobicity.

### 2.5. Differential Scanning Calorimetry (DSC)

The thermal properties of the samples were characterized according to the method by Zhao et al. [19]. A Q-2000 Differential Scanning Calorimeter (Discovery SDT650, TA Instruments, New Castel, DE, USA) was used to analyze the thermal stability of the samples. Twenty microliters of the sample solution was pipetted into an aluminum pan, sealed, and loaded into the calorimeter. An empty pan was used as the control. The test temperature range was from 20 to 150 °C with a heating rate of 10 °C/min. The temperature at the peak (T_p_) and the enthalpy change of denaturation (∆H) were determined.

### 2.6. Fourier Transform Infrared Spectroscopy (FTIR)

Modified according to the method by Qh et al. [20], the sample was mixed with KBr at a ratio of 1:100, ground, and pressed into flakes. The infrared spectrum of the sample was collected in a Vertex 70 infrared spectrophotometer (Bruker, Karlsruhe, Germany), the spectral range was 4000–400 cm^−1^, the test temperature was 25 °C, with a resolution of 4 cm^−1^, and a total of 64 scans.

### 2.7. Solution Stability Analysis via Multiple Light Scattering

The stability of eash solution sample was characterized using a multiple light scatterer (Turbriscan Tower, For-mulaction, Toulouse, France), with the vertical scan mode selected according to the instrument usage rules and the method reported by Du et al. [21]. An equal volume of sample solution was placed in a dedicated sample vial, and the sample was scanned using a pulsed near-infrared light source (λ = 880 nm). The optical signal was received every 40 μm, and a scan was performed at a temperature of 25 °C for 30 min, with a total detection time of 24 h. The Turbiscan stability index (TSI) reflected the change in signal intensity of transmitted and scattered light after the beam passed through a sample solution. An increase in the TSI value corresponds to the deterioration of sample stability [22], and the change in BS (ΔBS) was the BS value of the current sample minus the BS value of the first scans.

### 2.8. Foaming Capacity (FC) and Stability (FS)

The foaming property of the samples was determined with reference to the method by Watanabe, Shimada, and Arai [23]. Thirty milliliters of the complex solutions were placed into a 100 mL graduated cylinder and homogenized at a speed of 10,000 r/min for 2 min. The height of the foam above the liquid after mixing (V_0_) and after standing for 30 min at room temperature (V_30_) were recorded. The foaming capacity (FC) and foam stability (FS) values of the samples were calculated from the following formulas:(2)$$\mathrm{FC}\left(\%\right)=\frac{V_{0}-30}{30}\times 100\%$$
(3)$$\mathrm{FS}\left(\%\right)=\frac{V_{0}-30}{V_{30}-30}\times 100\%$$

### 2.9. Protein Solubility

Protein solubility was determined using the method by Perusko et al. [24]. First, the samples were diluted with PBS buffer to a protein concentration of 2 mg/mL in solution. The pH values of the sample solutions were adjusted to 3–9, respectively, and centrifuged at 5000× *g* for 30 min at 20 °C. The protein content of the supernatants was determined using the method by Lwory et al. [25] with BSA as the standard. Protein solubility was expressed as a percentage of the ratio of the protein content in the supernatant to the protein content in the original solution.

### 2.10. Statistical Analysis

All measurements were performed three times to ensure accurate results. The result were plotted using the Origin 2018 software (OriginLab Corporation, Northampton, MA, USA). Significant differences were determined by analysis of variance (ANOVA, St. Louis, MA, USA) using SPSS version 17.0 (SPSS Inc., Chicago, IL, USA) and defined as *p* < 0.05.

## 3. Results and Discussion

### 3.1. Effect of Polysaccharide Quality on Grafting Reaction

The glycosylation reaction, also known as the Maillard reaction, is a reaction between free amino groups of proteins and reduced carbonyl groups of polysaccharides. The glycosylation reaction of protein and polysaccharide molecules usually requires heat treatment. Ultrasound before heating can promote molecular movement between protein amino groups and polysaccharide carbonyl groups, providing good conditions for the reaction [13]. At the same time, it also reduces the formation of Maillard reaction intermediates [26,27]. The grafting degree is usually chosen to reflect the degree of the Maillard reaction [28]. Figure 1 shows the results of the grafting degree of the samples.

The grafting degree of PPI in the control sample without polysaccharide was recorded as 0%; the grafting degree of PPI and CSP of different concentrations under the same conditions were different. With an increase in the polysaccharide concentration, the grafting degree of the complexes gradually increased, and the grafting degree reached the highest when the mass ratio of PPI/CSP was 2:1 (38.57%); when the polysaccharide samples (PC1:1 and PC1:2) were added to graft, the branch size did not change significantly. Sonication can unfold protein structures and expose free amino groups in the protein structure [29]. When the system contained a small amount of polysaccharide carbonyl, the steric hindrance was small, the reactivity was high, and the glycosylation reaction rate was fast [7]; however, with an increase in the polysaccharide content, the change of grafting degree tended to be slower, and the glycosylation reached a saturated state. An increase in the polysaccharide addition led to a gradual increase in polysaccharide molecules in the solution and an increase in steric hindrance. The collision between the free amino group of the protein and the reducing end of the polysaccharide carbonyl group was inhibited, and the glycosylation rate was reduced.

### 3.2. Surface Hydrophobicity of PPI-CSP Complexes (H_0_)

Surface hydrophobicity (H_0_) is associated with protein-specific steric structural stability, and corresponds to hydrophobic interaction forces. This can lead to protein aggregation and precipitation, and can also characterize the number of hydrophobic sites between a protein surface and a polar solvent [30,31]. The effect of the polysaccharide addition on the surface hydrophobicity of the composite is shown in Figure 1. As compared with the control PPI, the surface hydrophobicity of PC 10:1 was significantly improved, but with the addition of polysaccharide, the surface hydrophobicity gradually decreased. During the glycosylation process, the cavitation effect produced by the ultrasonic treatment caused the protein structure to unfold, resulting in a large number of exposed hydrophobic groups. Although the polysaccharide contained a large number of hydrophilic hydroxyl groups, in the sample PC10:1, the added amount of polysaccharide was less, resulting in insufficient hydrophilic hydroxyl groups in the system, which improved the surface hydrophobicity of the complex [32]. However, the surface hydrophobicity of the complexes decreased significantly with increasing polysaccharide addition. The main reason for this could be that as the amount of polysaccharide added increased, the introduction of hydrophilic hydroxyl groups increased, which decreased the hydrophobic region on the protein, resulting in a decrease in the hydrophobicity of the complex [33].

There have been some similar conclusions obtained from a study on the surface hydrophobicity of soybean protein isolate-gum arabic conjugates [32]. Some scholars [34] have also put forward a similar view that as the number of polysaccharide molecules grafted with proteins increases, the surface hydrophobicity of proteins decreases rapidly. In the glycosylation reaction of polysaccharide and protein molecules, the surface hydrophobicity of the complexes are closely relate to the degree of grafting of the sample.

### 3.3. Thermal Properties

The properties of a protein are closely related to its structure. The glycosylation reaction can change the structure of protein, destroy some stable natural conformation forces of protein (such as hydrogen bonds, hydrophobic bonds, and disulfide bonds), and has an impact on the thermal properties of protein [35]. During food processing, proteins are easily denatured by heat. Improvement in the thermal stability of natural proteins has become of increasing interest in the industry. At present, the thermal properties of proteins and their complexes are mainly studied by DSC. Table 1 shows the thermal characteristic analysis results of each sample. T_d_ was the denaturation temperature. When the ambient temperature reached the denaturation temperature of the protein, the protein underwent thermal deformation, and therefore, the Td value reflected the thermal stability of the protein. The enthalpy value (∆H) of a sample is the energy change involved in the thermal transformation of a sample. Broken hydrogen bonds are endothermic (−∆H) and protein agglutination and hydrophobic reactions are exothermic (+∆H) [3].

Peanut protein isolate is mainly composed of conarachin and arachin. Therefore, the thermal stability of peanut protein and its complexes are related to conarachin and arachin. Referring to the results of Liu et al. [3], the denaturation temperature of conarachin is lower than that of arachin. From the thermal characteristic results in Table 1, it can be seen that peanut protein is denatured twice during the heating process, namely the thermal denaturation of conarachin (64.55 °C) and arachin (104.22 °C). The glycosylated samples still have two denaturation temperatures, and the T_d_ values are improved, with increasing polysaccharide addition. However, the denaturation temperature of arachin in the samples PC10:1 and PC5:1 do not change much; PC2:1, PC1:1, and PC1:2 also do not change significantly. In conclusion, the thermal denaturation temperature of the complexes increase with increasing polysaccharide addition. The increase in the Td value corresponds to an increase in the stability of the complex [36]. Therefore, the glycosylation method could improve the thermal stability of the protein, which was consistent with the research results of Ibanoglu et al. [37], Wang, Zhao, Yang, and Jiang [38], and Liu et al. [3]. During the glycosylation reaction, as the polysaccharide and protein molecules undergo the grafting reaction, the structure of peanut protein is changed, and the ability of the complexes to resist aggregation and sedimentation is enhanced. However, when the protein aggregates due to hydrophobic forces, the enthalpy value decreases. Therefore, with the enhancement of the ability of the complexes to resist aggregation, the enthalpy value required for the denaturation of the complexes gradually increases [39,40]. According to the research results of Liu et al. [3], when peanut protein undergoes glycosylation reaction, with an increase in reaction time, the Td value of peanut kernel increases, while ΔH gradually decreases. Excessive heat treatment destroys the structure of peanut kernels. Comparing this study with the conclusions of Liu et al. [3], it could also be seen that the glycosylation reaction conditions selected in this study did not show a reduction in the ∆H of the protein denaturation reaction caused by the heat treatment, and therefore, the reaction conditions were considered to be adequate.

### 3.4. FTIR Spectroscopy

FTIR spectroscopy provides information on the chemical composition and conformational structure of proteins through the absorption of radiation caused by vibrations between atoms in the molecule [4,41]. The FTIR spectra of the samples are shown in Figure 2. The absorption peak of the PPI-CSP complex was enhanced at 1650 cm^−1^, and this change was related to the C-N stretching vibration and N-H deformation vibration in the amide III band caused by the glycosylation process. This was consistent with the results of Su and others [42], who reported glycosylation of soy protein isolate with carboxymethyl cellulose. There was an obvious absorption peak in the wavelength range of 1300–1400 cm^−1^, which might be related to the stretching vibration of the CO bond in the CSP molecule [43]. The complexes also had a stronger absorption peak at 1043 cm^−1^. The glycosylation reaction of protein and polysaccharide molecules generate some new groups, such as C=O, C=N, and C-N, and the absorption peaks of these groups were in the range of 800–1800 cm^−1^ [15]. The appearance and enhancement of these absorption peaks indicate that the glycosylation reaction of PPI and CSP might generate new covalent bonds. However, the intensity of the absorption peak of the complex at 3398 cm^−1^ increased, which was mainly due to the -OH stretching vibration.

The change in the amide I band (1600–1700 cm^−1^) of protein FTIR was mainly the stretching vibration of the C=O bond, which reflected the secondary structure information of the protein [44]. The secondary structure was quantitatively analyzed by the second derivative infrared deconvolution spectral fitting method. The amide I band pattern was fitted using the PeakFit v4.12 software(Origin Lab Corp., Waltham, MA, USA) to obtain the percentage of each secondary structure. The frequency bands for each secondary structure were assigned as follows: 1610–1640 cm^−1^, 1640–1650 cm^−1^, 1650–1660 cm^−1^, and 1660–1700 cm^−1^ corresponding to *β*-sheet, random coil, *α*-Helix, and *β*-turns, respectively [45]. The results are shown in Table 2. After the glycosylation of PPI, the *α*-helix of the complexes were reduced, and the *β*-sheet, *β*-turns, and random coil were significantly increased (*p* < 0.05). However, when the PPI/CSP ratio reached 2:1, the secondary structure did not change significantly (*p* > 0.05). The secondary structure of PPI, shown in Table 2, changed after glycosylation. The reduction in *α*-helix was attributed to the binding of polysaccharide molecules to amino groups in the *α*-helix structure of proteins [46]. The glycosylation reaction transformed the ordered *α*-helical structure of PPI into *β*-sheet and random coil structure, which indicated that the covalent binding of high-added polysaccharide stretched the spatial structure of PPI, unfolded the peptide chain, and increased the flexibility [47]. Similar conclusions to the previous study were made by Li and Huang et al. [48] on the glycosylation reaction between peanut protein isolate and konjac.

### 3.5. Stability Analysis by Multiple Light Scatterer

Precipitation of a solution over time is due to thermodynamically incompatible repellency of protei and polysaccharide molecules. The effect of the mass ratio of PPI to CSP on the stability of the complex solutions was investigated using the multiple light scattering (MLS) technique. TSI and backscattered light intensity variation (ΔBS) can dynamically reflect global and local changes in system stability [49]. The larger the solute particles in a solution system, the higher the backscattered light intensity. When the solution systems were unstable and precipitation occurred gradually, the back astigmatism intensity increased at the bottom of the solution, the top of the solution gradually became clear, and the back astigmatism intensity gradually decreased. When the solutions were relatively stable, there was little change in the backscattered light intensity of the solution system with time (Appendix A) [50]. As shown in Figure 3, the slope of the average backscattered light intensity curved indicates the stability of the solutions [51].

With the addition of polysaccharide, the average value of the light intensity increased and the slope areas decreased. The back astigmatism intensity did not change much after PC2:1, which showed that the PC2:1 solution showed high stability. As reported in Table 3, with increasing polysaccharide addition, the BS value gradually decreased, and basically did not change after PC2:1; the TSI value became smaller and smaller, reaching the lowest value at PC2:1. This indicated that with the addition of polysaccharide, the smaller the intensity of backscattered light, the more extended the protein structure, and the more stable the solution state. When the amount of the polysaccharide reached a certain level, it became unstable again, which might be due to the agglomeration of the polysaccharide itself [52,53].

Therefore, combining the above characterization results of the PPI-CSP samples, we considered that PPI/CSP = 2:1 might be a critical concentration, implying that once this concentration was exceeded, thermodynamic incompatibility would drastically increase, which would lead to aggregation of protein structures and aggregation of polysaccharide molecules with themselves. The solution system became unstable, the intensity of backscattered light changed as the bottom of the solution increased, and precipitation gradually appeared macroscopically [54].

### 3.6. Foaming Properties

Foaming capacity (FC) is the ability of a protein to combine with gas to form a foam within a certain time, or the volume ratio of the foam to the initial protein solution. Foaming stability (FS) refers to the volume of water exuded from foam or the remaining volume of foam under certain conditions. When carbohydrates are complexed with proteins, they can act as thickeners or gelling agents to improve the foam stability of proteins [55].

As shown in Figure 4, the addition of CSP significantly improved the foamability and foam stability of the complexes, and it increased with an increase in the concentration of polysaccharide; the difference between samples was not significant when PC2:1 was reached. This might be related to the solubility of the complex. The polysaccharide complexation improved the solubility of PPI molecules and the softness of protein molecules. The dissolved PPI could be adsorbed on the surface of the foaming liquid film formed by liquid and bubbles to form protein foam [56]. When the solubility of PPI is high, it is easy to form an adhesive film between air and water, thereby, improving the FC of PPI. At the same time, the complexation of CSP inhibits the aggregation of PPI molecules, which is beneficial to improve the FS of the solution [54].

### 3.7. Protein Solubility at Different pH Conditions

Solubility can directly reflect the dispersion of protein in a solution, and can also reflect the stability of protein function. In the food industry, processing involves different acid-base changes, therefore, the study of the solubility of protein complexes under different acid-base conditions also has certain practical value. Through the characterization of the composite physicochemical and functional properties, the stability and functional activity of the sample PC2:1 were the optimum results, and therefore, this sample was selected for solubility characterization under different acid-base conditions. The results are shown in Figure 5; PPI had the lowest solubility at pH 5.

This was consistent with the research results of Li et al. [48]. When the pH value of the environment was near the isoelectric point of the protein, the surface charge of PPI in solution was close to zero, and the protein agglomerates and precipitates. The glycosylation reaction of PPI and CSP with a mass ratio of 2:1 could significantly improve the solubility of PPI, but did not change the isoelectric point of the complexes. This may be because, after the PPI was combined with CSP, the polysaccharide introduces a hydrophilic hydroxyl group, which improves the affinity of protein and water [12]. Ultrasonic treatment expanded the protein structure, and unfolding of protein structure can expose more amino groups to react with hydroxyl groups of polysaccharide molecules, thereby, reducing the hydrophobic area on the protein surface and improving the solubility of the complexes [57].

## 4. Conclusions

In this study, PPI-CSP complexes were obtained by glycosylation of peanut protein isolate with different amounts of corn silk polysaccharide using the ultrasonic-assisted moist heat method. The effects of different polysaccharide additions on the physicochemical and functional properties of the complexes were explored. With an increase in polysaccharide concentration, the grafting degree reached the highest at PC2:1. As compared with PPI, the complexes had higher thermal stability and lower surface hydrophobicity. The FTIR results proved that PPI and CSP formed glycosylated covalent complexes. The fitting analysis of the amide I band showed that the covalent binding of a high concentration of polysaccharide molecules stretched the spatial structure of PPI, expanded the peptide chain, and increased the flexibility. When the protein/polysaccharide ratio reached 2:1, the change was not significant. An analysis using a multiple light scattering instrument showed that the PC2:1 sample solution exhibited the highest stability. The foaming property and foam stability of PC2:1 were optimal, and the glycosylation treatment could significantly improve the solubility of PPI. This study explored the effect of polysaccharide addition on the glycosylation reaction of PPI, and provided a theoretical basis for the application of peanut protein isolate in the food industry.

## Figures and Tables

**Figure 1 foods-11-02214-f001:**
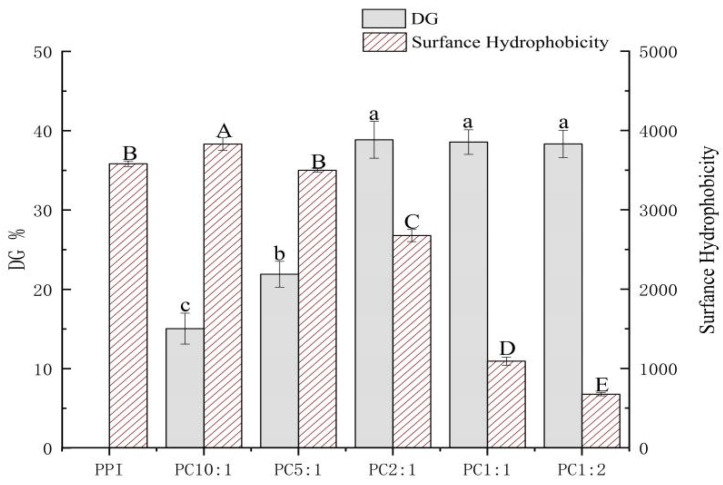
Degree of grafting (DG) and surface hydrophobicity (H_0_) of PPI-CSP complexes. The different lowercase or uppercase letters indicate that the results are significantly different (*p* < 0.05). The samples were peanut protein isolate, and the peanut protein isolate/polysaccharide ratios were 10:1, 5:1, 2:1, 1:1, and 1:2.

**Figure 2 foods-11-02214-f002:**
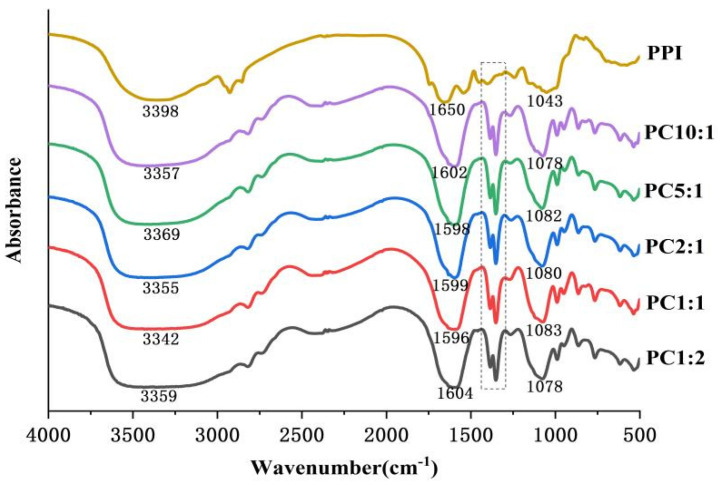
Fourier transform infrared spectra of complexes with different mass ratios of PPI and CSP. The dashed part is the new absorption peak generated by covalent binding.

**Figure 3 foods-11-02214-f003:**
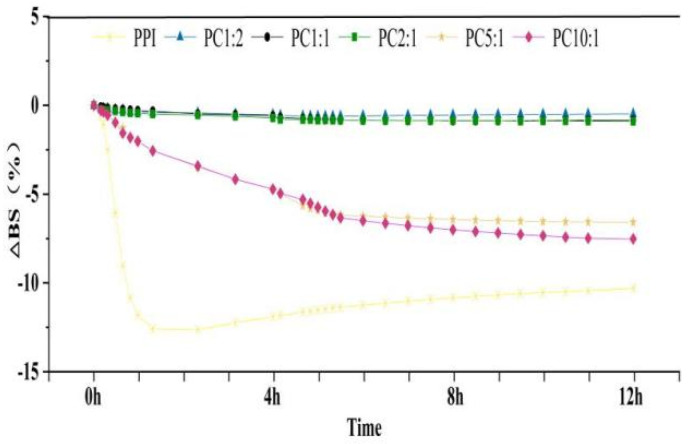
Average backscattered light intensity of the complex solutions in 12 h. The samples were peanut protein isolate, and the peanut protein isolate/polysaccharide ratios were 10:1, 5:1, 2:1, 1:1, and 1:2.

**Figure 4 foods-11-02214-f004:**
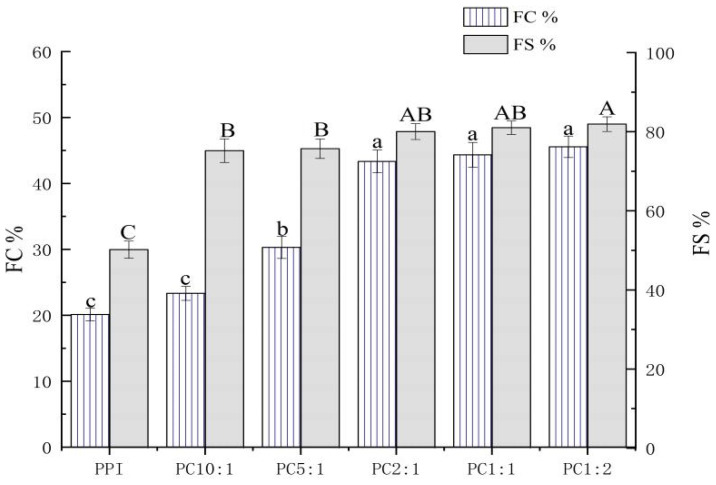
Foaming capacity and foam stability of the PPI-CSP complexes. Different letters in the graph represent significant differences (*p* < 0.05). The samples were peanut protein isolate, and the peanut protein isolate/polysaccharide ratios were 10:1, 5:1, 2:1, 1:1, and 1:2.

**Figure 5 foods-11-02214-f005:**
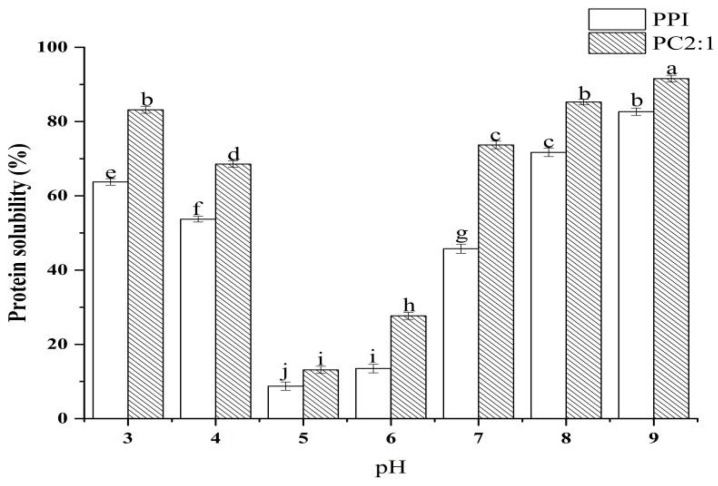
Changes in solubility of PPI and PC2:1 complexes with pH. Different letters in the graph represent significant differences (*p* < 0.05).

**Table 1 foods-11-02214-t001:** DSC characteristics of the PPI-CSP complexes.

Sample	Conarachin		Arachin	
	T_d_ (°C)	ΔH(J/g)	T_d_ (°C)	ΔH (J/g)
PPI	64.55 ± 0.23 ^e^	1221.6 ± 0.09 ^d^	104.22 ± 0.27 ^c^	281.7 ± 0.02 ^d^
PC10:1	69.83 ± 0.38 ^f^	1328.1 ± 0.10 ^c^	110.29 ± 0.31 ^b^	418.5 ± 0.08 ^c^
PC5:1	74.98 ± 0.21 ^d^	1430.3 ± 0.03 ^b^	110.31 ± 0.18 ^b^	486.1 ± 0.05 ^b^
PC2:1	86.06 ± 0.26 ^c^	1533.5 ± 0.07 ^a^	112.58 ± 0.24 ^a^	544.7 ± 0.13 ^a^
PC1:1	89.31 ± 0.15 ^b^	1553.7 ± 0.04 ^a^	112.43 ± 0.19 ^a^	543.3 ± 0.11 ^a^
PC1:2	90.56 ± 0.19 ^a^	1512.3 ± 0.06 ^a^	112.23 ± 0.32 ^a^	544.4 ± 0.14 ^a^

Means ± standard deviations of triplicate analyses are given. Superscript letters (^a–f^) indicate significant (*p* < 0.05) difference within the same column. The samples were peanut protein isolate, and the peanut protein isolate/polysaccharide ratios were 10:1, 5:1, 2:1, 1:1, and 1:2.

**Table 2 foods-11-02214-t002:** Secondary structure of complexes with different mass ratios of PPI and CSP.

Sample	*α*-Helix (%)	*β*-Sheet (%)	*β*-Turns (%)	Random Coil (%)
PPI	27.03 ± 0.13 ^a^	18.39 ± 0.27 ^d^	20.05 ± 0.15 ^d^	34.53 ± 0.12 ^d^
PC10:1	20.53 ± 0.21 ^b^	19.84 ± 0.19 ^c^	24.10 ± 0.24 ^c^	35.53 ± 0.17 ^c^
PC5:1	15.90 ± 0.18 ^c^	21.31 ± 0.23 ^b^	25.95 ± 0.11 ^b^	36.84 ± 0.06 ^b^
PC2:1	13.44 ± 0.29 ^d^	22.06 ± 0.08 ^a^	27.21 ± 0.34 ^a^	37.29 ± 0.25 ^a^
PC1:1	13.11 ± 0.05 ^d^	22.13 ± 0.12 ^a^	27.45 ± 0.15 ^a^	37.31 ± 0.06 ^a^
PC1:2	13.06 ± 0.34 ^d^	22.14 ± 0.16 ^a^	27.47 ± 0.09 ^a^	37.33 ± 0.14 ^a^

Means ± standard deviations of triplicate analyses are given. Superscript letters (^a–^^d^) indicate significant (*p* < 0.05) difference within the same column. The samples were peanut protein isolate, and the peanut protein isolate/polysaccharide ratios were 10:1, 5:1, 2:1, 1:1, and 1:2.

**Table 3 foods-11-02214-t003:** Turbiscan stability index (TSI) and backscattering intensity (BS) with different mass ratios of PPI and CSP compex solutions.

Sample	PPI	PC10:1	PC5:1	PC2:1	PC1:1	PC1:2
TSI	41.46	9.8	8.85	3.55	4.14	6.18
BS	16.88	12.3	12.04	9.27	9.26	9.25

The samples were peanut protein isolate, and the peanut protein isolate/polysaccharide ratios were 10:1, 5:1, 2:1, 1:1, and 1:2.

## Data Availability

No additional data were generated other than those reported in the manuscript.

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
