# Peer review of "Effects of Different Amounts of Corn Silk Polysaccharide on the Structure and Function of Peanut Protein Isolate Glycosylation Products"

_foods, 2022, doi:10.3390/foods11152214_

Round 1
Reviewer 1 Report
Han et al. have studied the effect of different amounts of corn silk polysaccharide on the structure and function of peanut protein isolate glycosylation products. The work has been done systematically and several physicochemical parameters essential for the study were carried out methodically. The results have been technically interpreted with scientific soundness. However, the main drawback of this paper is that the language presentation and it should be improved. In addition, the following points need to be addressed:
1. Abstract –PC2:1 condition should be described in words as ratio of protein to polysaccharide 2:1.
2. Section 2 - Please ensure that for all the materials, chemicals/reagents and instruments mentioned in section 2, company name, make and model along with city and country should be provided. For USA, in addition state name should be provided.
3. Equation 1-3 – the “100%” should be “100”.
4. Section 2.5 – The first sentence should be revised for clarity.
5. Section 2.9 – The first two sentences should be revised for clarity.
6. Figure 1, 3, 4 – in these figures the abbreviations used as axis labels should be provided in full form.
7. Figure 3 – the legends should be brought inside the figure. Also, the labels and values should be increased in size for clarity.
8. Table 1 – All of a sudden there is a usage of words conarachin and arachin in Table 1 and their spelling is different from their usage in subsequent text portions. These words should be explained as what in the footnote of this table.
9. Figure 2 – the FTIR band values should be increased in size for clarity.
10. Table 1 & section 3.5 heading – double check for typographical errors.
11. Section 3.6 – there is a discussion about isoelectric point, but not specified the value.
12. The number of references is too many and at least 10 references should be removed.
13. In all the tables ensure that the abbreviations used be provided in the full form in the respective table’s footnotes, while that used in figures be explained in the respective figure’s caption.
14. Hours and minutes should be abbreviated as “h and min” throughout the manuscript including tables and figures.
Author Response
Dear reviewers
Sincerely thanks for your suggestion. Those comments are all valuable and very helpful for revising and improving our paper, as well as the important guiding significance to our researches. We have studied comments carefully and have made correction which we hope meet with approval. The revised sections are highlighted with a yellow background in the paper. The responds to the reviewers’ comments are as following:
- Abstract –PC2:1 condition should be described in words as ratio of protein to polysaccharide 2:1.
Response: Thank you for your suggestions. We have changed PC2:1 in the manuscript to ratio of protein to polysaccharide 2:1.
- Section 2 - Please ensure that for all the materials, chemicals/reagents and instruments mentioned in section 2, company name, make and model along with city and country should be provided. For USA, in addition state name should be provided.
Response: Thank you for your suggestions. We have carefully checked and supplemented the reagent company, origin, country and other information.
- Equation 1-3 – the “100%” should be “100”.
Response: Thank you for your suggestions. We have corrected 100% to 100.
- Section 2.5 – The first sentence should be revised for clarity.
Response: Thank you for your suggestions. We have rewritten this section.
- Section 2.9 – The first two sentences should be revised for clarity.
Response: Thank you for your suggestions. We have rewritten this section.
- Figure 1, 3, 4 – in these figures the abbreviations used as axis labels should be provided in full form.
Response: Thank you for your suggestions. We have described each sample in detail in the figure caption.
- Figure 3 – the legends should be brought inside the figure. Also, the labels and values should be increased in size for clarity.
Response: Thank you for your suggestions. We have edited the figure.
- Table 1 – All of a sudden there is a usage of words conarachin and arachin in Table 1 and their spelling is different from their usage in subsequent text portions. These words should be explained as what in the footnote of this table.
Response: Thank you for your careful inspection, and we apologize for the inconvenience caused by our negligence. We have corrected the word usage in the table.
- Figure 2 – the FTIR band values should be increased in size for clarity.
Response: Thank you for your suggestions. We have edited the figure.
- Table 1 & section 3.5 heading – double check for typographical errors.
Response: Thank you for your suggestions. We are sorry to have troubled you due to our negligence. We have corrected the title.
- Section 3.6 – there is a discussion about isoelectric point, but not specified the value.
Response: Thank you for your suggestion. According to the existing literature and initial study results, the isoelectric point of peanut protein is near pH 5. This section mainly focuses on whether the complexes change the aggregation behavior of peanut protein. The specific isoelectric point value has no effect on the test results. Therefore, the specific value of its isoelectric point was not clearly indicated.
- The number of references is too many and at least 10 references should be removed.
Response: After our careful review, the references cited in the text all have their actual functions, therefore, the references have not been deleted.
- In all the tables ensure that the abbreviations used be provided in the full form in the respective table’s footnotes, while that used in figures be explained in the respective figure’s caption.
Response: Thanks for your suggestion, to make the research results clearer, we have added a detailed explanation of the sample nomenclature in the footnotes of the table and in the captions of the figures.
- Hours and minutes should be abbreviated as “h and min” throughout the manuscript including tables and figures.
Response: Thanks for your suggestion. We have abbreviated hours and minutes in the manuscript.

Reviewer 2 Report
The manuscript entitled, Effects of different amounts of corn silk polysaccharide on the structure and function of peanut protein isolate glycosylation products. The manuscript has novelty and contributes to the field. Below are comments/suggestions:
Line 18: what is PC
Lines 35-36: what about the allergy aspects of peanuts?
Line 64: will be ..? it should be past tense
All objectives should be in past-tense
Line 80: Shanghai Chian??
Line 111: revise the sentence. As such, it makes no sense
Line 124: A Q-2000 Differential Scanning Calorimeter (Discovery SDT650, TA Instruments, USA)… location of the company, which city?
Line 133: location of the company, city name
Line 180: there are no upper case letters
Grafting degree (DG) or GD? OR Degree of Grafting (DG)?
Figure 3. quality should be improved
None of the references are according to journal format. Please revise according to journal guidelines
Author Response
Dear reviewers
Sincerely thanks for your suggestion. Those comments are all valuable and very helpful for revising and improving our paper, as well as the important guiding significance to our researches. We have studied comments carefully and have made correction which we hope meet with approval. The revised sections are highlighted with a yellow background in the paper. The responds to the reviewers’ comments are as following:
Line 18: what is PC
Response: Thanks for the reminder, we have explained it in section 2.2 of the manuscript, PC is an abbreviation for the complex of peanut protein isolate and corn silk polysaccharide.
Lines 35-36: what about the allergy aspects of peanuts?
Response: Sincerely thanks for your suggestion. This study mainly focused on the effect of polysaccharides on the processing properties of peanut protein isolates, and did not study its allergenicity. Therefore, we did not discuss its sensitization, and we will follow your suggestion to conduct more in-depth research on its sensitization.
Line 64: will be ..? it should be past tense
All objectives should be in past-tense
Response: Thanks for your suggestion, we've changed that part to past-tense.
Line 80: Shanghai Chian??
Response: Thank you for your careful inspection. We have corrected Chian to China.
Line 111: revise the sentence. As such, it makes no sense
Response: Thanks for your suggestion. This part of the content is an explanation of the above formula, which has practical significance.
Line 124: A Q-2000 Differential Scanning Calorimeter (Discovery SDT650, TA Instruments, USA)… location of the company, which city?
Response: Thanks for your suggestion. We have supplemented the company location.
Line 133: location of the company, city name
Response: Thanks for your suggestion. We have supplemented the company location.
Line 180: there are no upper case letters
Grafting degree (DG) or GD? OR Degree of Grafting (DG)?
Response: Thanks for your suggestion, we have changed Grafting degree to Degree of Grafting.
Figure 3. quality should be improved
Response: Thanks for your suggestion. We have redone Figure 3.
None of the references are according to journal format. Please revise according to journal guidelines
Response: Thanks for your suggestion. The journal requires unification of the references, but no specific format, so we only unify the format of the references and provide the source of the references.
